# Endometrial Cancer in Reproductive-Aged Females: Etiology and Pathogenesis

**DOI:** 10.3390/biomedicines12040886

**Published:** 2024-04-17

**Authors:** Emma Bassette, Jennifer A. Ducie

**Affiliations:** 1Department of Obstetrics and Gynecology, Creighton University School of Medicine, Omaha, NE 68178, USA; emmabassette@creighton.edu; 2Methodist Gynecology Oncology, Nebraska Methodist Hospital, Omaha, NE 68114, USA

**Keywords:** endometrial cancer, premenopausal, reproductive-aged females, obesity, polycystic ovarian syndrome, nulliparity, lynch syndrome

## Abstract

Endometrial cancer is the most common gynecologic malignancy in developed countries, and the incidence is rising in premenopausal females. Type I EC is more common than Type II EC (80% vs. 20%) and is associated with a hyperestrogenic state. Estrogen unopposed by progesterone is considered to be the main driving factor in the pathogenesis of EC. Studies show that BMI > 30 kg/m^2^, prolonged duration of menses, nulliparity, presence of polycystic ovarian syndrome, and Lynch syndrome are the most common causes of EC in premenopausal women. Currently, there are no guidelines established to indicate premenopausal patients who should be screened. This review aims to synthesize current data on the etiology, risk factors, presentation, evaluation, and prognosis of endometrial cancer in this population.

## 1. Introduction

Endometrial cancer (EC) is the most common gynecologic malignancy in developed countries and is a well-established cause of morbidity and mortality in postmenopausal women, but the incidence has been steadily rising among premenopausal females [1]. The average age of diagnosis is the 6th decade of life, but there has been a 2.2% annual rise in diagnosis in women <50 years old [1]. An average of 20–25% of cases are premenopausal, but approximately 4% are in women <40 years old [2,3,4,5,6,7,8,9]. A study by Wan et al. (2016) found premenopausal EC cases were as high as 32.4% [10]. One study at Hospital Melaka in Malaysia reported that 44.8% of cases diagnosed were premenopausal women and as much as 16% of EC cases were diagnosed in women <40 years old [2]. In the United States, the incidence of EC in black women aged 35–49 years has increased by 2.8% per year from 2001 to 2019 [11].

Currently, the gold standard for EC treatment is surgical with total abdominal hysterectomy, bilateral salpingo-oophorectomy, and surgical staging. The increasing number of premenopausal women with this disease has led to the consideration of conservative management for those who are interested in future childbearing. In brief, these treatments include oral progestins with hysteroscopic ablation of lesions as well as long-acting levonorgestrel-releasing intrauterine devices (IUD) [12]. Women who meet the criteria for this treatment include those with low-grade endometrioid-type EC with no myometrial or lymphovascular invasion and no cervical or adnexal involvement [12]. A large study of 23,000 women with Stage IA EC had similar 5-year survival rates with progestin therapy when compared to standard treatment with hysterectomy [13]. Due to the low overall incidence of EC in premenopausal women, there are limited data regarding the most effective treatment regimen. A more detailed discussion of fertility-sparing treatment is out of the scope of this review.

According to the SEER Program database, women in the US have a 3.1% chance of developing EC [14]. Fortunately, the average 5-year survival rate of EC in all women is 81%, often reaching 100% in the premenopausal population, but timely diagnosis is important [6,14]. There is a growing body of evidence regarding etiologic factors leading to the development of EC in premenopausal women. This review aims to outline the current data on the incidence, etiology, evaluation, and prognosis of EC in premenopausal women.

## 2. Etiology and Risk Factors

Endometrial cancer can be divided into the following categories: Type I and Type II tumors. Type I tumors are FIGO Grade 1 endometrioid adenocarcinomas and are associated with unopposed estrogen stimulation. Type II tumors include FIGO Grade 3 endometrioid, serous, and clear cell carcinomas and are generally thought to be estrogen-independent. The incidence of Type I is 60–80% overall and that of Type II is 20% overall [15]. This paper focuses primarily on Type I EC, as it is much more common in premenopausal women. The most commonly reported risk factors for Type I EC are obesity, nulliparity, longer duration of menses (earlier age at menarche or late age of menopause), history of anovulatory cycles, polycystic ovarian syndrome (PCOS), diabetes mellitus (DM), or unopposed estrogen therapy. Hyperestrogenic states, or increased estrogen exposure, have been proven to be the main driver in the development of Type I EC.

### 2.1. Hyperestrogenic States or Estrogen Exposure

Studies have consistently demonstrated that premenopausal women with a hyperestrogenic state are at an increased risk of developing EC [8,9]. The hypothesis of the unopposed estrogen effect causing endometrial hyperplasia and neoplasia has been described since the 1970s [16,17]. The unopposed estrogen hypothesis, first described by Key et al. (1988), is a widely accepted theory stating that women with high circulating levels of endogenous estrogen unopposed by progesterone run an increased risk of developing EC [17]. This states that estrogen unopposed by sufficient progesterone leads to the unregulated mitogenic effects of estrogen, leading to increased endometrial proliferation during the follicular phase, and increased EC risk among young women with increased endogenous or exogenous estrogen levels [4,16,17]. This is in part due to adipose tissue with high concentrations of aromatase, converting androgens to endogenous estrogen. During the follicular phase, the release of estrogen alone without progesterone increases endometrial cell mitotic activity. Longer duration of unopposed estrogen exposure causing excess proliferation increases the probability of mutations, specifically in proto-oncogenes and tumor suppressor genes [16,17]. Progesterone is important in the luteal phase of the menstrual cycle to allow the secretory proliferation of the endometrial tissue [18]. Progesterone also plays a role in reducing mitotic activity by reducing estradiol (E2) receptor concentration in the endometrium as well as increasing the metabolism of E2 to estrone (E1), a less active estrogen [17].

Many large retrospective studies have shown that postmenopausal women with Type I EC have higher levels of circulating estrogens [19]. A study by Allen et al. (2008) examined the relationship between EC risk and serum levels of estrogens, androgens, and sex hormone-binding globulin (SHBG) in postmenopausal women [19]. They found significantly increased EC risk with the increased serum concentrations of total testosterone, free testosterone, estrone, total estradiol, and free estradiol. The ORs associated with BMI 25–29 and greater than or equal to 30+ were 1.26 and 2.67, respectively. Data in the premenopausal population continue to demonstrate results similar to these large population studies.

#### 2.1.1. Obesity

There is a rising rate of obesity in developed nations, which is a likely cause of the increase in the incidence of endometrial carcinoma in the premenopausal population. Obesity is established as a leading cause of Type I EC throughout the lifespan [20,21,22,23]. Of note, the incidence of Type II histology (clear cell, serous papillary) actually decreases with increasing BMI [24]. Higher BMIs have been associated with lower-grade cancers [9,22]. For example, Crosbie et al. (2012) found that WHO class III obesity (BMI > 40 kg/m^2^) was associated with a more significant proportion of isolated endometrial involvement and well-differentiated carcinomas when compared to normal-weight individuals (BMI 20–25 kg/m^2^) [24]. Other studies found that obesity (BMI > 30 kg/m^2^) is associated with a 4.5-fold increase in the incidence of EC compared to normal-weight women [25,26]. This relationship is thought to be due to changes in hormone metabolism, such as the endogenous production of estrogen through peripheral aromatization of androstenedione to estrone [16,21,27]. Obesity is also associated with decreased SHBG [17]. Prior to menopause, elevated E1 levels trigger a negative feedback loop that leads to oligo- or anovulation [27]. In patients with obesity, there is an increased basal level of E2 in circulation, stimulating the endometrial cells to undergo continuous proliferation that leads to endometrial carcinogenesis while never peaking to levels that induce ovulation, as discussed before. This increased E2 is likely not the main factor associated with the increased risk in obesity; it is the progesterone deficiency leading to unregulated endometrial proliferation [17].

The Million Women Study found that the relative risk of developing EC was 1.87 for every 5-kg/m^2^ increase in BMI in postmenopausal women [28]. A systematic review and meta-analysis by Renehan et al. (2008) demonstrated that, in premenopausal women, the risk of developing EC increases 1.59-fold with each 5 kg/m^2^ increase in BMI at a BMI of 28 and above [29]. In a study by Lachance et al. (2006), the average BMI of patients <45 yo was 40.3 kg/m^2^ compared to older patients (35.3 kg/m^2^ in 46–64 yo and 31.0 kg/m^2^ in those >65 yo) [7]. In a meta-analysis of BMI, the risk of EC increased more obviously when BMI was 25 or greater [23,30]. In one study by Pellerin & Finan (2005), 71% of patients <45 yo with EC had a BMI of 30 or greater [31]. In one study, the average BMI in the premenopausal group diagnosed with EC was 30.8 kg/m^2^ vs. 28.9 kg/m^2^ in the postmenopausal group [2]. Fortunately, while a higher BMI has been demonstrated to increase the incidence of EC and reduce the average age at diagnosis, it does not appear to lower survival rates [24].

One study by Nevadunsky et al. (2014) found that the age at diagnosis is decreasing linearly with increasing BMI [25]. They found that even when correcting for insulin resistance and hyperinsulinemia, factors that contribute to metabolic disorders, the linear association with BMI persisted. Grade and hypertension were also associated with age at the diagnosis of Type I EC. Interestingly, the presence of diabetes and hyperlipidemia were not significant when demonstrated as surrogate markers of metabolic disorders. This study did not stratify patients based on premenopausal vs. postmenopausal status, with the majority of patients being postmenopausal, but the association is interesting and may be further studied in premenopausal women. The BMI relationship appears to be linear, with more severe obesity (BMI > 40 kg/m^2^) potentially contributing to more pronounced carcinogenesis and an earlier development of dysplasia. Further investigation should analyze this linear relationship more closely and describe more specifically the implication of severe obesity in carcinogenesis and the development of dysplasia.


*ia. Comorbid Conditions*


Premenopausal women diagnosed with EC are often found to have comorbid conditions of diabetes and hypertension, though there are more robust data supporting the association of diabetes with EC. It is difficult to determine if these alone increase the risk or if they are associated as a result of obesity. In one study, hypertensive disease was more prevalent in non-endometrioid cancers, but there are significantly fewer data on hypertension alone to be associated with the development of EC [25].

Some hypotheses demonstrate endometrioid cancer to be associated with increased fasting insulin levels and insulin-like growth factors [22]. This may be due to hyperestrogenism rather than being an isolated cause, though Gunter et al. (2008) demonstrated that hyperinsulinemia may increase the risk independently from estradiol [22]. EC cell lines have also been shown to express high-affinity insulin receptors [32].

Chronic hyperinsulinemia is tumorigenic in estrogen-sensitive tissues [21]. This is due to the insulin-driven reduction of sex-hormone-binding globulin (SHBG), which increases estrogen availability [21,33]. Increased estradiol will stimulate endometrial cell proliferation and inhibit apoptosis, and it has been shown to stimulate insulin-like growth factor-I (IGF-I) in endometrial tissue [22,33,34]. IGF-I has a very similar amino acid sequence homology to insulin and has been found to have a stronger mitotic and antiapoptotic activity than insulin [34].

#### 2.1.2. Duration of Menses

Increased duration of the time of ovulation increases estrogen exposure in a woman’s life and has been linked to an increased risk of EC [16,17,35,36]. Reproductive factors that influence this lifetime exposure to estrogen are early age at menarche, late age of menopause, number of pregnancies, age at first pregnancy, and age at last pregnancy [18,35,37].

A review by Wu et al. (2019) analyzed 171 meta-analyses of 1354 individual studies of 53 risk factors and found that parity, higher age at last birth, and higher age at menarche were associated with a decreased risk of EC [37]. There is a 7–8% reduction in risk of developing EC per year of menstrual life, similar to the risk reduction associated with a shorter total menstrual lifespan HR 0.92 per year [35]. Furthermore, younger patients who present with EC are most commonly nulligravid with a history of infertility [3,38,39]. Women with a history of infertility may even have 2× increased risk of developing EC [38,39].

The age of menarche and menopause are also influenced by BMI. A study by Zhu et al. (2018) found that overweight and obese women run a 50% increased risk of late menopause, and underweight women are more likely to have early menopause [40]. Studies cited an age of menarche lower than 12 yo to confer an increased risk of EC [18,35,38]. A meta-analysis by Gong et al. (2015) found a modest 4% reduction in EC risk per 2 years of delay in menarcheal age [18]. They did estimate a 32% reduction in risk when the oldest menarcheal age was compared to the youngest, though the range of menarcheal ages varied amongst studies, ranging from 11 to 17 years [18].

Women with a later menopausal age have longer lifetime exposure to estrogen, which confers an increased risk of developing EC [20,37]. A study by Wu et al. (2019) aimed to summarize the evidence from observational studies to determine the association of age at menopause with the risk of EC [37]. This review explored eighteen epidemiologic studies: ten showed a significant association between later menopause and an increased risk of EC, while eight studies showed no association. The positive association between menopause and the development of EC appeared to be significant when the age of menopause was greater than 46.5 years old. The risk ratios (RR) for the development of EC were 1.17 (1.14–1.20), 1.57 (1.45–1.71), and 2.08 (1.80–2.39) when menopause occurred at 47, 50, 54, and 57 yo, respectively. It is thought to be due to later menopausal age associated with overweight BMI, later age of menarche, and higher parity.

The European Prospective Investigation into Cancer and Nutrition (EPIC) was a large, multicenter prospective cohort study designed to investigate the associations between nutritional, lifestyle, metabolic, and genetic risk factors, and cancer incidence. Dossus et al. (2009) utilized these data to examine the association of menstrual and reproductive characteristics with the development of endometrial cancer [35]. This study found the following characteristics of patients associated with EC when compared to age-matched controls: higher BMI, though this was not statistically significant [26.5 (5.4 SD) vs. 24.9 (4.4 SD)]; more often users of hormone replacement therapy (HRT) (44.3% vs. 39.8%); less likely to have achieved a university degree (18.4% vs. 24.1%); and slightly more often reported a history of diabetes (3.9% vs. 2.1%). The EC rate was lower among women with late menarche (defined as >15 yo) when compared to women with early menarche (defined as <12 yo) with HR 0.72 (0.58–0.90). They found an HR of 2.20 (1.61–3.01) for EC among women with late menopause (>55 yo) when compared to those with earlier menopause (<50 yo). The cause of this is still not fully understood, but the theory of unopposed estrogen is thought to be an important influence.

#### 2.1.3. Parity

A relationship between an increasing number of pregnancies and decreased EC risk has been infrequently studied. Many studies report nulliparous women to be at significantly higher risk of EC, but it is difficult to characterize the risk [38,41]. One study found an OR of 6.2 for nulliparous women and 5.9 for nulligravid women [38]. Higher age at last birth has been proposed to be associated with a lower risk of EC because decreased estrogen levels accompanying pregnancy among women approaching menopause may have a protective effect against cancer [35,42,43,44]. Evans-Metcalf et al. (1998) also found that nulliparity, rather than age, was found to be an independent risk factor for EC [45].

In a literature review by Gerli et al. (2014), parity showed strong evidence, delay in achieving menarche showed suggestive evidence, and prolonged breastfeeding showed weak evidence of a lower risk of EC [5]. The association of some factors such as age at first pregnancy or breastfeeding with EC risk has been controversial in prospective studies. Age at first delivery, hormone therapy use, and breastfeeding were not associated with EC risk [35]. A meta-analysis by Wang et al. (2015) attempted to assess the dose–response association between each breastfeeding month and EC risk [46]. In total, 14 studies were reviewed, including 5158 EC cases and 706,946 participants. Four studies reported a protective effect of breastfeeding while ten showed no significant association. The pooled risk ratio of developing EC and ever breastfeeding was 0.77. The RR of EC with ever breastfeeding vs. never breastfeeding was 0.85, though that association was not significant (95% CI 0.61–1.20), potentially due to classifying women who attempt breastfeeding once as having breastfed. The RR of the longest (2 years) vs. shortest (<1 month) category of breastfeeding was 0.71 (95% CI 0.53–0.95). Interestingly, when analyzing geographic locations, only studies conducted in North America a showed significant association between reduced EC risk and breastfeeding. The dose–response analysis found that increasing the duration of breastfeeding by one month led to a subtle 2%-decreased risk of EC (RR 0.98, CI 0.97–0.99).

Dossus et al. (2009) examined pregnancy-related variables in the EPIC cohort study [35]. This included the number of full-term pregnancies (FTP), age at the first and last FTP, and time since the last FTP. They found that parous women faced a decreased risk when compared to nulliparous women (HR 0.65, CI 0.54–0.77). When all pregnancy-related variables were examined together in the same model, only the number of FTP and time since the last FTP were significantly associated with decreased risk. An increasing number of FTP, defined as the sum of live-born children and stillbirths, conferred stronger protections (*p* = 0.04), and there was a decreased risk of EC associated with a cumulative duration of FTP (HR 0.78, CI 0.72–0.84 per year). They also found that the duration of breastfeeding was inversely associated with EC when comparing women who breastfed >18 months vs. those who breastfed <1 month (*p* = 0.01), though this was no longer significant when adjusting for the number of FTP (*p* = 0.22).

A study by Katagiri et al. (2023) examined 1005 EC cases using pooled individual data from 13 Asian prospective cohort studies conducted between 1963 and 2014 in the Asia Cohort Consortium [47]. This study found that participants who reported ever being pregnant faced a significantly lower risk of EC when compared to the nulliparous group (HR 0.54, CI 0.48–0.67). Specifically, hazard ratios of parous women when compared to nulliparous women were 0.54 in those with one–two deliveries, 0.50 in those with three–four deliveries, and 0.31 in those with five or more deliveries. This indicates a protective effect of childbearing against EC risk. In alignment with previous studies, the study also found that older age at menarche and younger age at menopause put women at a lower risk of EC. Ultimately, further investigation of these associations should be conducted to clarify the relationship of breastfeeding and parity as a protective factor, as data are contradictory regarding the significance of the effect on EC risk.

#### 2.1.4. Polycystic Ovarian Syndrome

Polycystic Ovarian Syndrome (PCOS) is relatively common, affecting up to 8% of women, and polycystic ovaries (PCO) without the clinical diagnosis of PCOS affect up to 20% of women [48]. The condition is characterized by oligo- or anovulation, signs of androgen excess, and an excess of small ovarian cysts [33,49]. Chronic anovulation leads to chronic estrogen exposure which, as stated above, can lead to endometrial hyperplasia and neoplasia. The association between PCOS and EC was described as early as 1949 and is based on several factors [50,51]. First, as discussed before, the stimulatory effect of estrogen on the endometrium when unopposed by progesterone has been shown to induce endometrial carcinogenesis [48,50,52,53]. Second, PCOS is associated with hyperinsulinemia and insulin resistance. Third, the hyperandrogenism associated with increased aromatase activity leads to the conversion of androgens to estrone, further leading to a hyperestrogenic state [54].

Other studies have described the presence of PCO without the presence of obesity being associated with EC [48,53]. In a systematic review and meta-analysis by Barry et al. (2014), patients with PCOS were at a significantly increased risk of EC (OR 2.79, 1.31–5.95) when comparing gynecologic cancers in women with PCOS <54 yo to age-matched controls [48]. Premenopausal patients with PCOS were found to be at a significantly higher risk of developing EC than postmenopausal patients with PCOS. This association did not change when correcting for BMI in PCOS vs. non-PCOS groups, suggesting that PCOS alone increases the risk of EC. Barry et al. (2014) also evaluated the risk of ovarian cancer in PCOS and found there was no significant increase overall, though Chittenden et al. (2009) did find an increased risk of ovarian cancer [48,49].

A systematic review by Chittenden et al. (2009) found that women with PCOS run up to three times the risk of developing EC when compared to the general population [49]. This included a study by Iatrakis et al. (2006), which found that the odds of developing EC in women with PCOS was OR 2.70 [55]. Iatrakis et al. (2006) also found that BMI, parity, a history of PCOS, diabetes, and the duration of menstrual cycles increased the risk of EC, with BMI being the most significant [55]. A study by Haoula et al. (2012) found a three-fold increase in EC risk in women with PCOS [52]. Wild et al. (2000) found that women with PCOS were six times more likely to develop EC [53]. A study by Gallup and Stock (1984) found that 31% of patients presenting with Type I EC had associated PCOS [56]. One case report describes adenocarcinoma in a 13-year-old girl with a BMI of 24.8, who was found to have polycystic ovaries [57]. A review of 10 cases of endometrial carcinoma in women aged 15–25 found that 70% exhibited the characteristics of PCOS clinically, with three of them having polycystic ovaries on diagnosis [58]. In this review, one patient had adenosquamous carcinoma that involved the ovary and pelvic sidewall; otherwise, the cancers were limited to the endometrium, which aligns with current data showing that the prognosis in these patients is very good.

Early treatment of PCOS may reduce the risk of the development of endometrial cancer by targeting BMI and insulin resistance. Insulin resistance is seen in 50–70% of women with PCOS [59]. Metformin has been demonstrated to decrease the inflammatory state associated with obesity in women with PCOS, primarily via the presence of increased inflammatory markers (increased CRP, IL-6, IL-8) [60]. In a study by Ali et al. (2019), which examined the effect of metformin vs. metformin and pioglitazone, a thiazolidinedione treatment for patients with PCOS, metformin use significantly decreased the level of serum fasting insulin (*p* < 0.001) [61]. When IL-8, a pro-inflammatory cytokine, was used as a marker for inflammation, a significant decrease was seen in patients with metformin and metformin and pioglitazone combination therapy.

Additionally, some authors have also reported that weight loss can decrease the risk of EC recurrence [62]. A meta-analysis by Graff et al. (2016) analyzed nine studies that evaluated the treatment effect of metformin compared to orlistat, a lipase inhibitor, in weight/BMI, HOMA score, testosterone levels, and insulin, and found a similar decrease in all measures, though none reached statical significance [63]. This meta-analysis was limited in that the studies lacked a substantial sample size and duration of treatment that may determine the impact of early treatment, though we may presume that it would be beneficial, as obesity is a known risk factor for the development of EC. Studies show that weight loss improves many factors, such as lipid profile and insulin resistance, and increases sex hormone-binding globulin (SHBG) concentration. Another single-center retrospective study by Sassin et al. (2023) evaluated metformin monotherapy versus metformin and tirzepatide (a GLP-1 and GIP receptor agonist) combination therapy [64]. They found that combination therapy doubled the chance of weight loss (HR = 2). There is significant heterogeneity in the interpretation of the BMI effect throughout studies, which makes it difficult to determine the specific risk of EC based on BMI alone.

It is still difficult to determine if PCOS alone increases the risk of EC because the underlying disease process is associated with several comorbidities: obesity, diabetes, metabolic syndrome, and anovulation leading to infertility. It is also made difficult due to the number of different accepted diagnostic criteria for the syndrome. All of the independently associated risk factors outlined above are seen in PCOS. There is a paucity of studies examining the relationship between PCOS and EC in a prospective longitudinal cohort, as would be the ideal evaluation of this association, and if treatment may prevent the development of EC. Due to the variation in diagnostic criteria for PCOS, associated risk factors, and a paucity of data evaluating the association between PCOS and EC, it is still difficult to discern the magnitude of risk increase for these patients. Prospective longitudinal cohort studies would be a useful tool to further investigate the relationships and stratification of risk based on specific comorbidities to evaluate risk profiles of individual variables.

#### 2.1.5. Combined Oral Contraception

Protection has been observed in women using combined oral contraception (OC), progestogens combined with estrogens, throughout their menstrual cycle [17,35,65]. Use for 1 year confers as much as 30–50% reduction in EC risk and extends for 10–20 years [6,35]. Dossus et al. (2009) found an HR of 0.65 (CI 0.56–0.75) among OC users, and the duration of use was inversely associated with risk (HR for >10 years 0.58, CI 0.42–0.79) when compared to those who used for one year or less (*p* < 0.0001) [35]. They also found that risk reduction by OC use was stronger among women with BMI > 30 when compared to those with BMI < 25 (*p* = 0.04). Further, lower postmenopausal endogenous estrogen concentrations have been observed in past OC users compared with never users [65].

Studies are contradictory; some report the use of hormonal contraception as increasing the risk of EC [19,38]. Andarieh et al. (2016) found that 41.5% of those who used contraceptive pills developed EC vs. 27.1% of users who did not [38]. Gitsch et al. (1995) found that 47% of women <45 with EC had a history of OCP use for two or more years [27]. More recent studies have been challenging the previous theory of HRT increasing the risk of EC [24,66] (Crosbie et al., 2012; Feinberg et al., 2019). Feinberg et al. (2019) did not find the use of HRT as a risk factor for the development of Type I or Type II EC [66].

### 2.2. Lynch Syndrome

Approximately 2–5% of endometrial cancers are associated with heritable predispositions [4,67]. Lynch syndrome accounts for the majority of inherited endometrial cancers and 6% of all endometrial cancers [67]. Lynch syndrome is an autosomal dominant disease caused by mutations in DNA mismatch repair genes *hMLH1*, *hMSH2*, *hMSH6*, and *hPMS2*, and can cause cancers of the endometrium, renal pelvis, ovary, stomach, small bowel, and ureter [67]. The majority of Lynch-associated endometrial cancers are FIGO Type I [67]. Type I cancers have been shown to be associated with sporadic deletions in k-Ras, PTEN, or mismatch repair mechanisms [66]. EC is actually the most common extracolonic manifestation of Lynch syndrome. The risk is approximately 40–60% in mutation carriers and can be up to 71% with the specific mutation *hMSH6* [67]. These patients often develop EC 10 years prior to the average age of sporadic EC [67]. In a study by Boks et al. (2002), 92% of Lynch syndrome-associated cancers were Type I compared to 88% in the control group [68].

Identifying premenopausal women with EC has important implications for both the patients and their families. Furthermore, it is not uncommon to present with a synchronous diagnosis of gynecologic malignancy and colorectal malignancy [69]. In 2007, the Society of Gynecologic Oncology published guidelines for genetic risk assessment for Lynch Syndrome in patients with endometrial or colorectal cancer [38,70]. They recommended a genetic risk assessment for patients that ran a 20–25% increased risk: (1) met the Amsterdam criteria, (2) had synchronous or metachronous endometrial OR ovarian and colorectal cancer diagnosed <50 yo, or (3) had 1st or 2nd degree-relative OR personal history of colorectal or EC with the evidence of mismatch repair defect. The Amsterdam criteria are as follows: three or more relatives with Lynch-associated cancer; one of the relatives should be a first-degree relative of the other two, cancers should involve two successive generations, and one cancer should be diagnosed <50 yo [67]. Patients with a greater than 5–10% risk of having inherited predisposition are as follows: (1) those with endometrial OR colorectal cancer <50 yo; (2) those with synchronous or metachronous Lynch-associated tumor at any age; (3) those diagnosed with endometrial OR colorectal cancer at any age with two or more first- or second-degree relatives with Lynch-associated tumors at any age; or (4) patients with a first- or second-degree relative that meets any of the previous criteria. The guidelines report that a genetic risk assessment may be helpful, but do not necessarily recommend it in these patients. Soliman et al. (2005) found that only 3% of women with synchronous endometrial and ovarian cancers met the aforementioned Amsterdam Criteria for genetic evaluation and concluded that it is unlikely for Lynch syndrome to account for a significant amount of young EC cases [9]. Some studies have proposed using low BMI as a screening tool for Lynch syndrome in young women with endometrial carcinoma [9]. Many studies show that a family history of reproductive cancers increases the risk of early presentation of EC and as such is an important component of workup in premenopausal patients with EC [9,38].

There are minimal data available at this time regarding any significant differences in morbidity or mortality in women with Lynch-associated EC. According to the SEER database, mortality rates did not increase in patients <45 yo who underwent ovarian preservation with Stage I EC [71]. Boks et al. (2002) found that the 5-year survival rate was 88% vs. 82% for Lynch-associated and sporadic endometrial cancers, respectively [68].

## 3. Presentation

In postmenopausal women, the presentation of EC or hyperplasia is much more straightforward, with post-menopausal bleeding. EC in premenopausal women presents with persistent abnormal uterine bleeding, which can present as menorrhagia (heavy menstrual cycles), prolonged menstrual bleeding, or irregular cycles [2,72]. The presentation in fertile women is more convoluted, given that abnormal uterine bleeding affects up to one-third of reproductive-aged women, which often leads to delayed diagnosis [2,45,73]. There is no defined difference in the presentation of benign vs. premalignant abnormal uterine bleeding in the premenopausal patient [2]. In one study, the average duration of symptoms in premenopausal women was three times longer than in postmenopausal women [45]. One study in Malaysia found that the median time of diagnosis for premenopausal women was 12 weeks, but some women were not diagnosed for up to 2 years [2]. Despite the delay in presentation and diagnosis, younger women often have early-stage disease [45]. Most studies report a higher percentage of Type I endometrial carcinomas, up to 90%, but others show a rate of Type II carcinomas that is no different from postmenopausal women [45]. A recent study at Parkland Hospital evaluated the percentage of endometrial pathology in women who presented to the women’s emergency department for AUB [74]. They found that 17.4% of patients had intermediate pathology and 2.1% had endometrial malignancy. The mean BMI was 36.6 kg/m^2^. It is important to be aware that patients may present initially to an emergency department with heavy vaginal bleeding requiring same-day evaluation.

The above-outlined risk factors for EC ultimately increase the rate of endometrial proliferation and hyperplasia. Endometrial hyperplasia, defined as endometrial thickening with proliferation of irregularly sized and shaped glands and an increased gland-to-stroma ratio, is the only known precursor for invasive disease. Endometrial intraepithelial neoplasia (EIN) is a contemporary nomenclature used to define two clinical categories of hyperplasia: (1) normal polyclonal endometrium diffusely responding to an abnormal hormonal environment, and (2) intrinsically proliferative monoclonal lesions that arise focally and confer an elevated risk of adenocarcinoma [6]. Endometrial hyperplasia is further characterized by the presence or absence of atypia, which occurs when there is nuclear atypia of the endometrial gland cells [6]. Women with atypical endometrial hyperplasia run up to 43% risk of concurrent EC, which highlights the need for screening guidelines [75].

Pennant et al. (2017) conducted a systematic review to evaluate the risk of EC and atypical hyperplasia in premenopausal women with abnormal uterine bleeding [76]. They stratified AUB into heavy menstrual bleeding (HMB) and intermenstrual bleeding (IMB) and found that the risk of EC was higher in women with IMB (0.52%) when compared to HMB (0.11%). Overall, they found only a 1.33% risk of EC in premenopausal women with AUB, though they did not stratify subgroups based on other risk factors, only menstrual status.

### Concurrent Ovarian Malignancies

Synchronous endometrial and ovarian carcinoma (SEOC) is a very rare cause of primary double cancer. SEOC accounts for 40–51.7% of the overall incidence of double gynecologic cancers of 0.63–1.7% [77,78,79]. The risk factors for SEOC are very similar to those for EC in premenopausal women. Premenopausal age, obesity, infertility and decreased parity, and Lynch syndrome [79] Soliman et al. (2004) found that the average age at SEOC diagnosis was 50 years and >50% occurred in premenopausal women [80]. The Scully Criteria, first published in 1998, were 12 criteria to differentiate between metastasis and double primary cancers that considered histologic features, the depth of invasion and tumor extension, ovarian tumor location, laterality, and the depth of invasion and degree of spread [81]. The Scully criteria are still widely used but have been slightly modified in recent years to distinguish metastatic cancer from primary double cancer better, though these new modifications are rarely used due to the small sample size of the initial study [82]. The theory behind the presence of SEOC is called “microenvironment confinement”, meaning that tumor cells detach from the primary lesion and spread spatially without undergoing apoptosis rather than the usual route of EC metastasis through lymphatic, hematogenous, or implantation [79]. The overall incidence of SEOC is very rare and needs a more robust study to better inform the risk of ovarian conservation for fertility preservation in premenopausal women.

Ovarian involvement has been evaluated in several studies of premenopausal EC risk. The studies vary significantly in regard to the reported rate of concurrent ovarian malignancies, ranging between 5% and 29% [27,78,83,84]. In one study, 13% of patients <40 yo with Lynch syndrome and endometrial carcinoma had synchronous ovarian tumors [85]. Gitsch et al. (1995) found that a significantly higher percentage of cases were associated with simultaneous ovarian malignancies in patients younger than 45 when compared to those >45 years (29.4% vs. 4.6%) [27]. This is important to consider when deciding to proceed with ovarian conservation, as the risk of concurrent ovarian carcinoma is 2–7 times that of postmenopausal women [41,83].

Evans-Metcalf et al. (1998) found that premenopausal women are five times as likely to present with a concurrent ovarian malignancy [45]. The tumors in the premenopausal group were all Stage I and all but one were evident grossly, though previous studies, albeit dated, reported a significantly larger portion of occult disease [86]. This warrants further study, as fertility conservation is often a goal in premenopausal women diagnosed with EC. Ovarian tumors that are unilateral, large in size, without surface involvement or multinodular growth pattern and the presence of endometriosis or borderline tumor are more likely to be an independent primary tumor [77]. Of course, it is imperative to diagnose these pathologically.

Sun et al. (2013) found that 5.4% had gross ovarian involvement, but only 1.5% were a synchronous ovarian malignancy rather than metastasis [84]. They found that adnexal metastasis was more likely when associated with lymph node (LN) involvement, >50% myometrial invasion, and gross extrauterine disease. Interestingly, tumor histology, grade, and cervical involvement did not increase the risk of adnexal metastasis. In patients without intraoperative evidence of ovarian involvement, only 1.2% had pathology-confirmed synchronous malignancy. Overall, no difference was found in prognosis or survival between those who underwent BSO compared to those who kept one or both ovaries in situ. The authors also conducted a meta-analysis that was consistent with their findings.

Walsh et al. (2005) studied women 24–45 yo who underwent hysterectomy for EC [78]. They found coexisting epithelial ovarian tumors in 25% (n = 26) of the study population, a majority of those being synchronous primaries (n = 23). Only one had mucinous ovarian cancer; the remaining tumors were of epithelial origin. 71% of patients had abnormal adnexal imaging prior to surgery, 29% had benign/normal radiologic findings, 67% had ovarian masses discovered intraoperatively, 14% had ovarian surface abnormalities, and 19% (n = 4) had no intraoperative abnormalities. Of the four cases with no intraoperative findings, three had operative reports commenting on cystic but normal-appearing ovaries. Lee et al. (2007) reported a synchronous ovarian malignancy rate of 7.3%, but only 0.97% of cases were without intraoperative extrauterine disease [83].

More recent studies have shown that ovarian preservation does not worsen prognosis [84]. These data are reassuring when making treatment decisions for fertility preservation, but there remains a paucity of data on long-term outcomes of ovarian preservation from an oncologic standpoint and those must be further studied so that clinicians can make more informed decisions regarding the risks and benefits of ovarian preservation.

## 4. Evaluation

The presentation of women with EC has been outlined previously, and it is apparent that the indication for evaluation is not straightforward. It is well known that transvaginal sonography showing endometrial stripe greater than or equal to 4 mm in postmenopausal women warrants further workup, but what is the appropriate indication for workup in premenopausal women? One study recommended that women younger than 45 years old with chronic excess estrogen exposure, failed medical management, and persistent abnormal uterine bleeding should undergo endometrial sampling [76]. Another study suggested that women 35 years and older with recurrent anovulation, women under 35 years with risk factors for EC, and those unresponsive to medical therapy should undergo endometrial biopsy [72]. Ultimately, young women with abnormal uterine bleeding and risk factors such as nulliparity, PCOS, diabetes, and high BMI should be referred to a licensed provider for endometrial biopsy [2].

The evaluation and workup for EC are not benign, and it is not cost-effective to evaluate every patient with abnormal uterine bleeding (AUB) given that the diagnostic yield is very low and such a small percentage of premenopausal women are diagnosed with EC [76]. Experts have attempted to create endometrial thickness guidelines for pre-menopausal women, but this proves difficult due to significant variation in normal menstruation. Studies have suggested ranges from 4 mm to 16 mm, which makes diagnostic accuracy very low [87,88,89]. Endometrial biopsy using pipelle vs. dilation and curettage are both acceptable as first-line evaluation.

The UK NICE guidelines recommend endometrial biopsy at the time of hysteroscopic evaluation for women with persistent intermenstrual, irregular, or heavy menstrual bleeding who are obese or have PCOS, women taking tamoxifen, or for women whose treatment for HMB has been unsuccessful [90]. They do not recommend endometrial biopsy (EMB) without hysteroscopic evaluation and do not specify the length of medical management that qualifies the patient as failed. Ultimately, there are no well-defined guidelines for calculating a patient’s overall risk and it is currently up to clinical judgment without specific parameters. Further research should be conducted to inform the development of evidence-based screening guidelines.

With the diagnosis of endometrial cancer, molecular classification provides vital prognostic information that may inform patients and providers about expected disease behavior and oncologic outcomes. The Cancer Genome Atlas (TCGA) identified four distinct molecular classifications for endometrial cancer [91]. Micro RNAs, or miRNAs or miRs, are single-stranded, non-coding RNAs that modulate gene expression. miRs have been shown to impact the cell cycle through the regulation of metabolic pathways and can influence both oncogenic and tumor-suppressive activity. It has been proposed that we may use miRs as a promising biomarker for the detection and diagnosis of endometrial cancer. And, depending on the miR identified, one may also be able to determine the biologic behavior and prognosis of a particular cancer, which may play a significant role when thoroughly counseling a patient who desires future fertility about therapeutic options for their newly diagnosed endometrial cancer [92].Much more research needs to be conducted to identify biomarkers that will best assist in the diagnosis of endometrial cancer, as well as provide prognostic information that may guide conversations between patients and their providers with respect to the best treatment for their cancers and the safety and likelihood of preserving fertility.

## 5. Prognosis

Fortunately, young women with EC tend to present with an early stage and favorable histologic subtype. The majority of endometrial cancers diagnosed in premenopausal women are Type I, which is associated with a hyperestrogenic state, factors as outlined above [2,7]. High BMI correlates with low tumor grade, endometrioid histology, and early stage of presentation [77]. According to Fadhlaoui et al. (2010), 70% of endometrial adenocarcinomas are Stage I and 90% are Grade 1 [4]. Studies report that the frequency of early-stage Grade 1 tumors reaches up to 90% in the premenopausal population [6,58] A majority of EC cases in premenopausal women are in the early stage, but Gitsch et al. (1995) reported 29% of premenopausal cases were Stage III and IV [27]. Similarly to the post-menopausal group, the FIGO stage, grade, histological type, and degree of myometrial invasion are associated with a worse prognosis [41].

A study by Tran et al. (2000) found that the presence of myometrial invasion was more than 50% lower in young women (24%) when compared to women >45 years old (49%) [93]. This study also noted that for the most part, young patients had an equivalent distribution of pathologic features compared to postmenopausal women. There was a statistically significant progression-free survival in the premenopausal group as well; only 1.5% developed recurrence within 1 year compared to 6% in the postmenopausal group. In the premenopausal group, the risk of recurrence in 3 yo was 3% vs. postmenopausal of 10%; and the 5-year recurrence rate remained unchanged, while it increased to 12% in the postmenopausal group [27]. Lee et al. (2007) reported only a 2.4% risk of recurrence [83]. Overall, the average 5-year survival for young women with EC is approximately 93% [31,82].

A retrospective single-center study by Biler et al. (2017) studied 40 women <40 yo with surgically treated EC [41]. The average age of diagnosis was 38 years old, 85% of tumors were early stage, 67.5% were low-grade, and 97.5% were endometrioid; 72.5% displayed <50% myometrial invasion. These numbers are similar to other studies [84]. Notably, CA 125 level >35 was the only statistically significant prognostic factor for both shorter progression-free survival and overall survival.

Type II EC patients are more likely to be non-white and have lower BMI, a postmenopausal status, a history of other malignancy, and higher gravidity and parity than those with Type I cancers [66]. Type II, nonendometrioid endometrial cancers, are much more frequent in young women with Lynch syndrome when compared to those without, though Lynch syndrome usually presents with Type I cancers [67,77]. Undifferentiated and dedifferentiated carcinomas are often associated with Lynch syndrome or other sporadic mismatch repair abnormalities [77].

In the rare cases when patients present with SEOC, age plays an important role in prognosis. Premenopausal patients younger than 50 yo have been found to have a 94.1% 5-year survival rate and decreased risk of recurrence compared to postmenopausal women, who have a 53.7% 5-year survival rate [94].

In a cross-sectional study by Evans-Metcalf et al., (1998) it was found that women <45 years old were more likely to have a low-grade disease (65% vs. 38%), more likely to have synchronous ovarian malignancy (11% vs. 2%), and be nulliparous, and that overall survival was no different between women <45 yo when compared to those >45 yo. Women <45 yo had a higher median weight and were twice as likely to be nulliparous than women >45 yo. There was no statistically significant difference in personal or first-degree relatives affected by breast, colon, or endometrial malignancies. Interestingly, this study found that the prevalence of Stage I disease was not significantly different between pre- and postmenopausal women, and the occurrence of endometrioid, adenosquamous, serous, and clear cell tumors was equal. In women <45 yo, the disease was more likely to be confined to the endometrium (Stage Ia) compared to women >45 yo.

## 6. Conclusions

There is evidence that the incidence of EC in the premenopausal population continues to rise. The aforementioned literature supports the etiology of this increase, but evidence-based recommendations for screening young women for EC still do not exist. The most commonly reported risk factors are obesity, nulliparity, longer duration of menses (earlier age at menarche or late age of menopause), a history of anovulatory cycles, PCOS, diabetes mellitus, and/or unopposed estrogen therapy. Many of the studies demonstrated that BMI remained directly correlated with increased EC risk when correcting for other comorbid conditions, but hyperestrogenic conditions such as PCOS and Type II diabetes mellitus seem to compound that risk when also present.

There are gaps in the current literature and limitations to the outlined studies. While there are several case reports on adolescent females, very few data exist regarding EC in this population [57]. With the rise of the obesity epidemic in the pediatric population, it is imperative to recognize adolescent females who are at an increased risk of developing estrogen-driven EC earlier in life. Interventions in this population may prevent the continued rise in EC incidence. Some of the larger cohort studies did not stratify patients based on menopausal status, which is why it is important to analyze more accurately the unique risk factors present in premenopausal women and how they change throughout the lifespan.

EC is the leading cause of gynecologic cancers in the US and other developed nations, but there is a paucity of data in developing and underdeveloped nations [38]. This limits the generalizability of many studies, given the differences in lifestyle, genetic factors, access to healthcare, and the availability of cancer treatment among many others. Additionally, much of the population data are not disaggregated into ethnicity, which is crucial to a deeper understanding of the environmental and epigenetic risk factors. Additionally, a disparity in the funding of EC research was made apparent by Katcher et al. (2024) [95]. Using the SEER database to determine the burden of disease in women’s cancers and the National Cancer Institute (NCI) website for research funding, they demonstrated that uterine cancer had the lowest funding despite the rising incidence. These inequities observed in cancer funding are significantly affecting vulnerable populations. Large efforts should be made to advocate for increased funding for uterine cancers so that we can provide equitable care.

Providers should consider the factors outlined in this review when deciding which reproductive-aged patients should be evaluated for EC. It is important for clinicians to consider all patient factors that confer a higher risk of EC. These risk factors include BMI, the age of menarche lower than 12 years, the age of menopause greater than 46.5 years, PCO and PCOS, nulliparity and a number of FTPs, concurrent genetic predispositions, and family history. We encourage the community to pursue large prospective cohort studies that disaggregate data based on race and ethnicity as well as premenopausal status so that we can continue to better understand the increasing incidence of EC in premenopausal women.

This review has aimed to outline the current literature regarding the development of endometrial cancer in premenopausal women. Overall, the population of premenopausal women with endometrial cancer is small, though incidence continues to rise, making further study with broader population studies and more robust longitudinal data imperative for a better understanding of the complex interplay of factors influencing EC development and to inform evidence-based guidelines for approach to evaluation in this population.

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
