# Peer review of "Endometrial Cancer in Reproductive-Aged Females: Etiology and Pathogenesis"

_biomedicines, 2024, doi:10.3390/biomedicines12040886_

Round 1

Reviewer 1 Report

Comments and Suggestions for Authors

I read with great interest the Manuscript titled “Endometrial cancer in reproductive aged women females: Etiology and Pathogenesis”

 which falls within the aim of Biomedicines.

In my honest opinion, the topic is interesting enough to attract the readers’ attention. Methodology is accurate and conclusions are supported by the literature review. Nevertheless, authors should clarify some point and improve the discussion citing relevant and novel key articles about the topic.

- What are the actual clinical implications of this review compared to previous ones? it is important to report the summarized results obtained by the authors in the context of clinical practice and to adequately highlight what contribution this review adds to the literature already existing on the topic and to future study perspectives.

- Please explain how the use of metformin in PCOS could be also helpful in reducing endometrial cancer risk (see PMID: 31683341)

- Introduction may benefit from at least a brief paragraph mentioning the fertility-sparing approach, which is extremely useful in childbearing-age women with IA-B endometrial cancer (see doi: 10.1155/2022/4070368.)

Author Response

I read with great interest the Manuscript titled “Endometrial cancer in reproductive aged women females: Etiology and Pathogenesis”.

which falls within the aim of Biomedicines.

In my honest opinion, the topic is interesting enough to attract the readers’ attention. Methodology is accurate and conclusions are supported by the literature review. Nevertheless, authors should clarify some points and improve the discussion citing relevant and novel key articles about the topic.

- What are the actual clinical implications of this review compared to previous ones? It is important to report the summarized results obtained by the authors in the context of clinical practice and to adequately highlight what contribution this review adds to the literature already existing on the topic and to future study perspectives.

This is discussed in the conclusion section on page 13.

- Please explain how the use of metformin in PCOS could be also helpful in reducing endometrial cancer risk (see PMID: 31683341)

Thank you, please see page 7, lines 313 to 337.

- Introduction may benefit from at least a brief paragraph mentioning the fertility-sparing approach, which is extremely useful in childbearing-age women with IA-B endometrial cancer (see doi: 10.1155/2022/4070368.)

We have included a brief paragraph discussing. I believe there will be another article discussing those in more detail. See line 35 to 47.

Reviewer 2 Report

Comments and Suggestions for Authors

Endometrial Cancer in Reproductive Aged Females: Etiology and Pathogenesis.

Reviewer statement:

Endometrial cancer is the most common gynecologic malignancy in developed countries, a well-established cause of morbidity and mortality in postmenopausal women. The incidence of endometrial cancer is rising in premenopausal women. There is growing evidence regarding etiologic factors leading to development of endometrial cancer in premenopausal women. The authors conducted a review aiming to outline the current data in incidence, etiology, evaluation, and prognosis of endometrial cancer in premenopausal women which is excellent and clinical relevant. This information could help in daily practice and could guide further research.

Title: The title chosen reflect the study being reported and is considered adequate.

Overall:

As a reader it was a pleasure to read the article; it was easy to read and to understand. The English grammar was adequate. The authors should be complimented for this achievement.

Abstract Should be adjusted based on the provided comments

Introduction  

The introduction section is well written. The background as a reason for performing the current study is well explained. No comments on this section.

Etiology and risk factors

This section is also well written, describing the  etiology and risk factor of endometrial cancer. The presentation helps the reader understanding etiology and risk factors. Thereby , also discussing research gaps, which is excellent.  No  comments on this section.

Presentation

This section is also well written, explaining symptoms of endometrial cancer especially in premenopausal women. The presentation section is very useful.

Only one minor point, the abbreviation LN in line 446 on page 9 should be written out completely as this is not familiar to all readers

Evaluation

This section is well written and good to understand. The evaluation and workup  for endometrial cancer are well discussed.  

Only one minor point, the abbreviation AUB in line 484 and EMB in line 195 on page 10 should be written out completely as they are not familiar to all readers

Prognosis

This section is  well written, discussing the prognosis of endometrial cancer patients. No comments on this section.

Conclusion

No comments on this section

Author Response

Endometrial cancer is the most common gynecologic malignancy in developed countries, a well-established cause of morbidity and mortality in postmenopausal women. The incidence of endometrial cancer is rising in premenopausal women. There is growing evidence regarding etiologic factors leading to development of endometrial cancer in premenopausal women. The authors conducted a review aiming to outline the current data in incidence, etiology, evaluation, and prognosis of endometrial cancer in premenopausal women which is excellent and clinically relevant. This information could help in daily practice and could guide further research.

Title: The title chosen reflects the study being reported and is considered adequate.

Overall:

As a reader it was a pleasure to read the article; it was easy to read and to understand. The English grammar was adequate. The authors should be complimented for this achievement.

Abstract Should be adjusted based on the provided comments

Introduction  

The introduction section is well written. The background as a reason for performing the current study is well explained. No comments on this section.

Etiology and risk factors

This section is also well written, describing the  etiology and risk factor of endometrial cancer. The presentation helps the reader understanding etiology and risk factors. Thereby, also discussing research gaps, which is excellent.  No comments on this section.

Presentation

This section is also well written, explaining symptoms of endometrial cancer especially in premenopausal women. The presentation section is very useful.

Only one minor point, the abbreviation LN in line 446 on page 9 should be written out completely as this is not familiar to all readers.

Author response: Thank you, we have made this change.

Evaluation

This section is well written and good to understand. The evaluation and workup  for endometrial cancer are well discussed.  

Only one minor point, the abbreviation AUB in line 484 and EMB in line 195 on page 10 should be written out completely as they are not familiar to all readers

Author response: Thank you, we have made these changes.

Prognosis

This section is  well written, discussing the prognosis of endometrial cancer patients. No comments on this section.

Conclusion

No comments on this section

Reviewer 3 Report

Comments and Suggestions for Authors

The article "Endometrial Cancer in Reproductive Age: Fertility-Sparing Approach, Reproductive Factors and Endometrial Cancer Risk Among Women, Reproductive Factors and the Risk of Endometrial Cancer, Risk Factors for endometrial carcinoma among postmenopausal women, Age as an Independent Predictor of outcome in endometrial cancer" reviews various studies to explore the etiology and pathogenesis of endometrial cancer (EC), mainly focusing on reproductive-aged women.

The limitations of the studies are as follows:

 1. Generalizability and Population Specificity - The findings from studies conducted in specific geographic locations, such as North America, may need to be more generalizable to populations in other parts of the world due to differences in lifestyle, genetic predispositions, and healthcare systems.

 2. Methodological Limitations - Some studies did not stratify patients based on premenopausal vs. postmenopausal status, which is crucial for understanding the impact of various risk factors on EC risk across different life stages. The reliance on observational studies and meta-analyses introduces potential biases such as recall bias, selection bias, and confounding factors, which can affect the accuracy of the findings.

 3. Controversial and Conflicting Evidence - Associations between certain reproductive factors (e.g., age at first pregnancy, breastfeeding) and EC risk have been controversial, with prospective studies showing conflicting results. This indicates a need for further research to clarify these associations.

 4. Comorbid Conditions and Causality - It is challenging to determine whether comorbid conditions like diabetes and hypertension independently increase EC risk or if their association is primarily due to obesity. This complexity underscores the need for studies that can dissect these relationships more clearly.

 5. Specificity of Associations - The association between polycystic ovarian syndrome (PCOS) and EC risk is well-established. Still, the studies reviewed often do not differentiate between the effects of PCOS itself and the comorbid conditions commonly associated with PCOS, such as obesity and insulin resistance.

 6. Lack of Longitudinal Data - There is a noted lack of prospective longitudinal cohort studies explicitly examining the relationship between PCOS and EC, which would provide more definitive evidence regarding causality and the magnitude of risk.

 7. Variation in Diagnostic Criteria - The variation in diagnostic criteria for PCOS across studies makes it difficult to compare results and draw firm conclusions about the association between PCOS and EC risk.

 8. Influence of Body Mass Index (BMI) - The relationship between BMI and age at natural menopause and the linear association between BMI and age at diagnosis of Type I EC highlights the significant role of obesity in EC risk. However, the linear relationship and its implications for carcinogenesis and dysplasia development in more severe obesity cases require further investigation.

In summary, while the article provides valuable insights into the risk factors and associations with endometrial cancer, the limitations of the reviewed studies highlight the need for more rigorous research methodologies, broader population studies, and longitudinal data to understand better the complex interplay of factors influencing EC risk.

Comments on the Quality of English Language

MINOR

Author Response

The article "Endometrial Cancer in Reproductive Age: Fertility-Sparing Approach, Reproductive Factors and Endometrial Cancer Risk Among Women, Reproductive Factors and the Risk of Endometrial Cancer, Risk Factors for endometrial carcinoma among postmenopausal women, Age as an Independent Predictor of outcome in endometrial cancer" reviews various studies to explore the etiology and pathogenesis of endometrial cancer (EC), mainly focusing on reproductive-aged women.

The limitations of the studies are as follows:

  1. Generalizability and Population Specificity - The findings from studies conducted in specific geographic locations, such as North America, may need to be more generalizable to populations in other parts of the world due to differences in lifestyle, genetic predispositions, and healthcare systems.

Author response: Thank you for these comments and those below. You can find response to this on page 13.  

  1. Methodological Limitations - Some studies did not stratify patients based on premenopausal vs. postmenopausal status, which is crucial for understanding the impact of various risk factors on EC risk across different life stages. The reliance on observational studies and meta-analyses introduces potential biases such as recall bias, selection bias, and confounding factors, which can affect the accuracy of the findings.

Author response: This can be found on 621-624.

  1. Controversial and Conflicting Evidence - Associations between certain reproductive factors (e.g., age at first pregnancy, breastfeeding) and EC risk have been controversial, with prospective studies showing conflicting results. This indicates a need for further research to clarify these associations.

Author response: You can find these comments on lines 268 to 271.

  1. Comorbid Conditions and Causality - It is challenging to determine whether comorbid conditions like diabetes and hypertension independently increase EC risk or if their association is primarily due to obesity. This complexity underscores the need for studies that can dissect these relationships more clearly.

  1. Specificity of Associations - The association between polycystic ovarian syndrome (PCOS) and EC risk is well-established. Still, the studies reviewed often do not differentiate between the effects of PCOS itself and the comorbid conditions commonly associated with PCOS, such as obesity and insulin resistance.

  1. Lack of Longitudinal Data - There is a noted lack of prospective longitudinal cohort studies explicitly examining the relationship between PCOS and EC, which would provide more definitive evidence regarding causality and the magnitude of risk.

  1. Variation in Diagnostic Criteria - The variation in diagnostic criteria for PCOS across studies makes it difficult to compare results and draw firm conclusions about the association between PCOS and EC risk.

Author response: In response to comments 4-7: This is discussed on page 7.   

  1. Influence of Body Mass Index (BMI) - The relationship between BMI and age at natural menopause and the linear association between BMI and age at diagnosis of Type I EC highlights the significant role of obesity in EC risk. However, the linear relationship and its implications for carcinogenesis and dysplasia development in more severe obesity cases require further investigation.

Author response: Response can be found on page 3, lines 144 to 146.

In summary, while the article provides valuable insights into the risk factors and associations with endometrial cancer, the limitations of the reviewed studies highlight the need for more rigorous research methodologies, broader population studies, and longitudinal data to understand better the complex interplay of factors influencing EC risk.

Round 2

Reviewer 3 Report

Comments and Suggestions for Authors

thank you for the revision.

i am satisfied with revisions.

Author Response

Thank you for reviewing our manuscript.